# The effect of intramuscular injection technique on injection associated pain; a systematic review and meta-analysis

**Oluseyi Ayinde** [ORCID]*◉, **Rachel S. Hayward**◉, **Jonathan D. C. Ross**

Whittall Street Clinic, University Hospitals Birmingham NHS Trust, Birmingham, United Kingdom

◉ These authors contributed equally to this work.
* oluseyi.ayinde@uhb.nhs.uk

**Data Availability Statement:** All relevant data are within the paper and its Supporting Information files.

**Funding:** The authors received no specific funding for this work.

## Abstract

### Aim

To review the effect of different intramuscular injection (IMI) techniques on injection associated pain, in adults.

### Methods

The review protocol was registered on PROSPERO (CRD42019136097). MEDLINE, EMBASE, British Nursing Index and CINAHL were searched up to June 2020. Included studies were appraised and a meta-analysis, where appropriate, was conducted with a random effects model and test for heterogeneity. Standardised mean difference (SMD) with a 95% confidence interval in reported injection pain (intervention cf. control) was reported.

### Results

29 studies were included in the systematic review and 20 studies in the meta-analysis. 13 IMI techniques were identified. 10 studies applied local pressure to the injection site. Of these, applying manual pressure (4 studies, SMD = -0.85[-1.36,-0.33]) and Helfer (rhythmic) tapping (3 studies, SMD = -2.95[-5.51,-0.39]) to the injection site reduced injection pain, whereas the use of a plastic device to apply local pressure to the skin (ShotBlocker) did not significantly reduce pain (2 studies, SMD = -0.51[-1.58,0.56]). Acupressure techniques which mostly involved applying sustained pressure followed by intermittent pressure (tapping) to acupressure points local to the injection site reduced pain (4 studies: SMD = -1.62 [-2.80,-0.44]), as did injections to the ventrogluteal site compared to the dorsogluteal site (2 studies, SMD = -0.43[-0.81,-0.06]). There was insufficient evidence on the benefits of the 'Z track technique' (2 studies, SMD = -0.20[-0.41,0.01]) and the cold needle technique (2 studies, SMD = -0.73[-1.83,0.37]) on injection pain. The effect of changing the needle after drawing up the injectate on injection pain was conflicting and warming the injectate did not reduce pain. Limitations included considerable heterogeneity, poor reporting of randomisation, and possible bias in outcome measures from unblinding of assessors or participants.

## Conclusions

Manual pressure or rhythmic tapping over the injection site and applying local pressure around the injection site reduced IMI pain. However, there was very high unexplained heterogeneity between studies and risk of significant bias within small studies.

## Introduction

Intramuscular injection (IMI) is a common clinical procedure with 16 billion injections administered globally every year [1] in a wide variety of healthcare settings [2]. The widespread use of IMI reflects the range of medications that can be delivered via this route including sedatives, hormonal therapies, vaccines, tumour immunotherapy, immune-suppressants, long-acting antipsychotics, vitamins and antibiotics [3, 4]. The intramuscular route offers a potential advantage by improving drug absorption and bioavailability compared to oral and other parenteral routes [4, 5], and can be used when the tolerability of oral medication is poor [6], and to ensure treatment adherence [4].

Injection site pain is common following IMI [7–9]. Anxiety and fear associated with pain can reduce the acceptability of treatment to patients [10], and for clinicians the knowledge that a procedure is painful may reduce its use. Given the frequency of IMIs and their importance as a treatment option, IMI site pain is an important issue and a number of pharmacological [7], psychological [11], and procedural (injection technique) interventions [12] have been proposed to reduce injection associated pain. Pharmaceutical interventions, such as injectable or topical anaesthetics can reduce pain [7], but are not always compatible with the medication being injected and may be associated with drug side-effects, allergies and increased cost [13]. Physical and procedural interventions, through the use of an optimal injection technique, have the potential to reduce pain while having little effect on the length or cost of the procedure.

Previous interventional and observational studies have evaluated IMI techniques which reduce pain with many focused on childhood vaccinations. In this younger population pain can be decreased by having the child sit up (or by holding an infant), stroking the skin or applying pressure close to the injection site before and during injection, and performing a rapid intramuscular injection without initial aspiration [14]. A recent systematic review of IMI techniques in adults suggests that IMI to the ventrogluteal site, the Z track, and manual pressure IMI techniques may be effective interventions in reducing IMI associated pain [15]. However, this review restricted the range of electronic databases searched and the language of studies included, and may have missed relevant studies. Given the importance of IMI pain in the management of patients, we provide an extended and updated review of the current evidence on IMI techniques used to reduce pain using a robust search and clear inclusion criteria. The aim of this study was to systematically review the effect of different intramuscular injection techniques on injection pain in adults.

## Methods

A systematic review protocol was developed and registered with PROSPERO at the Centre for Reviews and Dissemination, University of York (Registration No. CRD42019136097 http://www.crd.york.ac.uk/PROSPERO/display_record.php?ID=CRD42019136097).

### Eligibility criteria

Studies were considered eligible for review if they met the following inclusion criteria; involved human participants; male or female; aged ≥ 18 years (or the majority of the study population

was aged ≥ 18 years); involved IMI of any medicinal or non-medicinal product; specified an injection technique to reduce injection site pain; had a comparator group that also received an IMI of any medicinal or non-medicinal product; and collected data on pain secondary to IMI. There was no restriction on the indication for treatment, the health status of participants, or the healthcare setting. There was also no restriction on the language of publication or the publication date.

## Search strategy

Scoping searches were initially carried out to refine the search strategy. Thereafter, the OVID search platform was used to search MEDLINE (1946 to 29th of June 2020) and EMBASE (1974 to 29th of June 2020). The British Nursing Index and Cumulative Index Nursing and Allied Health Literature (CINAHL) were searched via EBSCOhost (1981 to 29th of June 2020). The search terms are shown in S7 Table. Citation searching was carried out on included articles.

## Study selection

All identified records were entered into Endnote X7 and duplicates removed. Titles and abstracts, where available, were screened independently by two reviewers (RH and OA) for relevance using the inclusion criteria. Full text articles were sought for all potentially relevant records and the inclusion criteria were applied independently by the same two reviewers. Any disagreement was resolved by discussion or by a third independent reviewer (JR). Foreign language records were included when searching, and titles and abstracts were translated to allow screening. All potentially relevant foreign language studies were translated (full text) for assessment and, if appropriate, data extraction. Reviewers were not blinded to the authors or settings of the studies.

## Data extraction

The data extraction form was designed and piloted on three included studies, and finalised following the pilot. Data extraction was performed independently by two reviewers (RH and OA) on all included studies. Disagreements were resolved by discussion and by a third independent reviewer (JR). The following study characteristics were collected; (i) study author; (ii) study design; (iii) country of publication; (iv) number of participants; (v) age range of participants; (vi) gender of participants; and (vii) ethnicity of participants. Specific details of the intramuscular injection technique used in the intervention and comparator groups were collected: (i) clinical setting; (ii) definition/description of the intramuscular injection technique used; (iii) indication for IMI; (iv) substance administered; (v) volume of injection; (vi) needle gauge used; (vii) needle length used; (viii) intramuscular site of injection; (ix) presence of co-morbidities; (x) frequency of injection and; (xi) length of follow up. Data on the outcome measure (pain secondary to IMI) including: (i) definition/description of pain measurement tool used (ii) timing of pain assessment (iii) information on who completed the pain assessment (iv) severity of pain (v) duration of pain (vi) summary statistics, and test statistics.

## Risk of bias

Risk of bias assessment was included within the data extraction form and was independently assessed by two reviewers (RH and OA). The risk of bias of randomised controlled trials (RCTs) was assessed with the Cochrane Collaboration's tool for assessing risk of bias of RCTs [16], for quasi-experimental studies this was performed with the non-randomised studies on interventions (ROBINS-1) tool [17], and for systematic reviews this was assessed with the risk of bias in systematic reviews (ROBIS) tool [18].

## Meta-analysis

Meta-analysis was performed for the primary outcome when appropriate and possible. Studies using either parallel or crossover designs were combined in the meta-analysis. Results from both periods of crossover studies were used unless there was reason to believe carryover of effects from one period to another posed a serious problem. IMI pain was considered as a continuous outcome, and in studies with multiple intervention or comparator (control) arms, the interventions (Raddadi et al. [19]) or control (Çelik and Korshid [20]) arms were combined and pooled means and standard deviations were estimated. Standardised mean differences (SMD) were used to estimate the outcome, since pain was measured with a variety of scales between studies. In studies where pain was measured with multiple instruments, preference was given to measures from the 'visual analogue scale' (VAS), then the 'numerical rating scale' (NRS) and the 'verbal rating scale' (VRS) in that order. In studies where pain was measured at different time points post injection (Kanika and Rani [21]; Khanra et al. [22]), the outcome measure at the earliest reported time was used in the meta-analysis. Studies with no discernible control arm were excluded from the meta-analyses.

**Statistical details.** In studies with crossover design, within-study comparisons were based on paired t-tests. Correlations (rho) between repeat outcomes on the same patient were estimated when possible from P-values, paired t-statistic or from any relevant summary data. When correlations could not be estimated, they were imputed for each outcome using the lowest (positive) estimate among other studies in the meta-analysis [23, 24]. Sensitivity analyses were undertaken to investigate the robustness of results to imputed quantities.

For parallel arm studies the SMD was calculated as the difference in mean outcome between groups divided by the standard deviation of outcome among participants, and for cross-over studies the SMD was calculated as the mean of the within patient difference between the intervention and comparator injection pain outcome, divided by the between patient standard deviation of the outcome. An approximate variance for the SMD for crossover studies was taken as, $\sqrt{2(1 - rho)}$ [25] where rho is the estimated or imputed correlation between repeated outcome measurements. Not all studies reported the same outcome measures (e.g., means, standard deviations), therefore some of the effect sizes were calculated with transformed data. An SMD of zero means that the intervention and the comparator have equivalent effects on IMI pain, and SMDs lower than zero indicate that the intervention was more beneficial than the comparator.

**Heterogeneity and subgroup analyses.** Clinical heterogeneity was examined prior to performing a meta-analysis but did not preclude the combination of results. Studies were also found to be methodologically heterogeneous and thus random-effects analyses with the DerSimonian-Laird method were undertaken in preference to fixed effect analyses in order to encompass residual variation between studies into the confidence interval for a pooled effect. Sensitivity analyses were undertaken to investigate the impact of this preference on the estimate. $I^2$ was used as an indicator of heterogeneity, $I^2$ value of 0% indicates an absence of dispersion, and larger values show increasing levels of heterogeneity [26]. Random effects meta-regression was used to compare subgroups where appropriate. The following subgroup analyses were specified *a priori* and were carried out:

i. The intervention effect where IMI techniques have similar operating procedures

ii. The effect of experimental design (RCT vs Quasi-experimental studies) on the pooled estimate

**Reporting bias.** Funnel plots were used to assess the risk of reporting bias (small study and/or publication bias). Contour enhanced funnel plots and Egger's meta-regression test for funnel plot asymmetry were conducted.

**Sensitivity analyses.** The following sensitivity analyses were carried out to assess the robustness of our findings:

i. Fixed effect meta-analyses (which do not incorporate heterogeneity between studies)

ii. Analyses that ignore crossover design of studies

iii. Exclusion of small studies from the meta-analysis of interventions using local pressure techniques—parallel studies with study size $\leq 100$ and cross over studies with studies size $\leq 50$ were excluded.

iv. Inclusion of the Najafidolatbad et al. [27] study into the Z track meta-analysis. This study did not have a control arm as a comparator but rather compared two interventions.

v. Analyses that ignore the L14 pressure point (other acupressure studies were conducted with the UB/BL32 or UB31 pressure point) intervention arm in the Raddadi et al. [19] study.

Statistical analyses were performed with Stata 15 for windows, and a two tailed P<0.05 was considered statistically significant.

# Results

Database searches identified 604 potential articles. Duplicates were removed and two reviewers screened 397 articles independently. 355 records were excluded including 5 studies where full texts were not available. 42 articles and a further 13 studies identified from citation searches were independently assessed for eligibility against the pre-specified inclusion criteria by the same two independent reviewers. Any disagreements were resolved by discussion. 29 studies met the inclusion criteria and were included in the qualitative analysis and 20 studies were included in the quantitative analysis. The PRISMA flow diagram in Fig 1 shows the study selection process.

## Characteristics of included studies

Table 1 summarises the characteristics of the included studies. 29 studies, with a total of 2442 participants, comprising 13 randomised controlled trials (RCTs), 15 quasi-experimental studies and 1 systematic review/meta-analysis were included in this descriptive analysis. The sample size of the experimental studies ranged from n = 25 to n = 242. Where gender distribution was reported, the study participants were mostly women (1431/2186). The clinical setting for the studies varied, with 14 of the 28 (50%) experimental studies conducted with hospital inpatients. Thirteen (manual pressure, Helfer skin tap, ShotBlocker, post injection massage, Z track, acupressure, two-needle technique, altering injection speed, cold needle, application of ice to the injection site, airlock technique, gluteal injection site and altering the temperature of the injectate) different IMI techniques to reduce injection pain were evaluated within the included studies. In studies reporting 'standard technique' as the control, the definition of 'standard' varied considerably and included insertion of the injecting needle at $90^0$ to the skin [20], aspiration prior to injection [20], a two needle technique (changing the needle between drawing up material and injecting it) [28] and the airlock technique (including a small quantity of air in the syringe) [29–31]. However, in other studies, aspiration [29, 30, 32], the airlock technique [32, 33], and $90^o$ needle insertion [32–36] were used as part of the intervention. 27/28 of the experimental studies used a quantitative scale as their outcome assessment measure —a visual analogue scale was the most common (19/28), followed by a numerical rating scale (5/28). One study [37] used a 4-point Likert scale, utilising the descriptors "None, Mild, Moderate, or Severe".

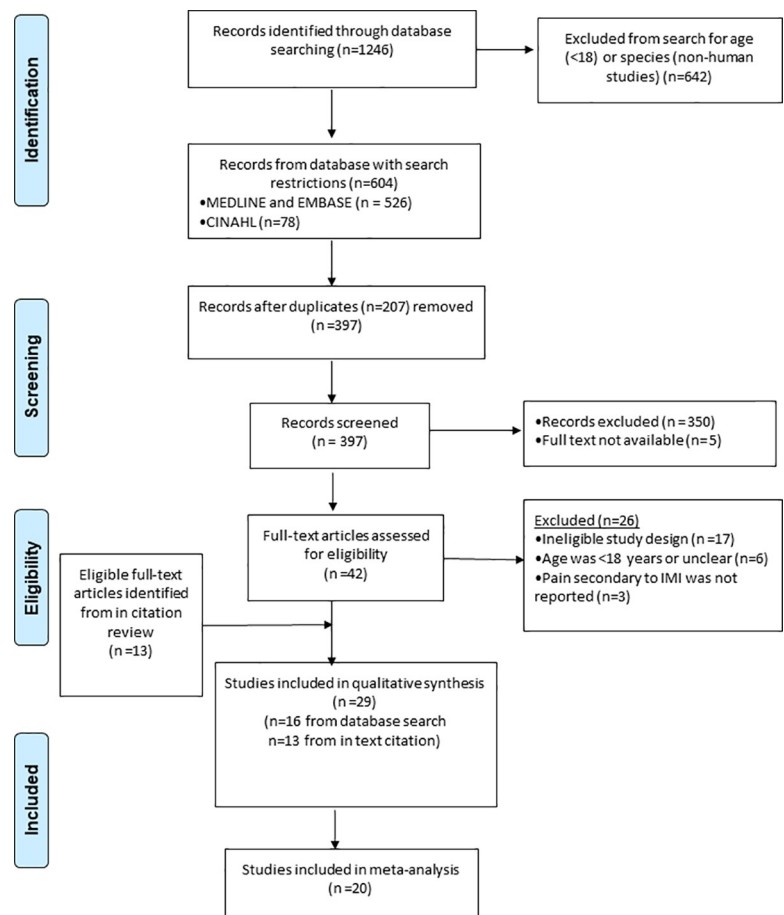

**Fig 1. PRISMA flow diagram for the systematic review of intramuscular injection technique and the effect on pain.**

## Risk of bias assessment

Studies included in this review were of variable methodological quality. The reporting of sequence generation and allocation concealment were often unclear in the included RCTs. Pain is a subjective measure and blinding of participants was reported in 4/13 (31%) of RCTs (Fig 2). The overall risk of bias for the quasi-experimental studies was low or moderate, with a moderate risk of bias associated with unblinding in the 'measurement of the outcome' domain for majority (9/15 [60%]) of the studies (Table 2). For the included systematic review and meta-analysis, the overall risk of bias was low, but the risk of bias in the 'study eligibility criteria' domain was high (Table 2).

## IMI techniques reporting significant reduction in injection pain

In Table 3, we report the mean pain scores for IMI interventions and their respective comparators. Manual pressure (4 studies), Helfer skin tap (3 studies), post injection massage (1 study), acupressure to UB31/UB32/BL32 acupoint (4 studies), airlock technique (1 study), and application of ice to the injection site (1 study) IMI techniques were found to be potentially beneficial in reducing injection pain compared to their respective comparators.

All four quasi-experimental studies that employed the application of manual pressure to the injection site reported a significant reduction in injection pain [28, 34, 38, 39]. Three of these

**Table 1. Characteristics of included studies.**

| Study (Year of publication) | Design | Country | Sample Size | Gender | Age Range* (years) | Population | Intervention to reduce pain | Comparator | Pain Measurement Tool | Timing of pain assessment |
|---|---|---|---|---|---|---|---|---|---|---|
| Barnhill et al. [34] 1996 | Quasi-experimental | USA | 93 | M32 F61 | >18 | Community | Manual pressure | No pressure | Visual Analogue Scale | Immediately after injection |
| Chung et al. [38] 2002 | Quasi-experimental | Hong Kong | 74 | M33 F41 | 18–42 | Community | Manual pressure | Standard technique | Verbal Rating Scale | After each injection |
| Zore and Dias [39] 2014 | Quasi-experimental (crossover design) | India | 50 | M20 F30 | 15–55 | Hospital outpatient | Manual pressure & progressive muscle relaxation therapy | Standard protocol (undefined) | Numerical Rating Scale and Behavioural Observation | Following injection |
| Öztürk et al. [28] 2016 | Quasi-experimental | Turkey | 123 | M13 F110 | 18–30 | Hospital outpatient | Manual pressure | Standard technique | Numerical Rating Scale | Immediately after injection |
| Shah and Narayan [13] 2016 | Quasi-experimental (crossover design) | India | 82 | M39 F43 | 20–60 | Hospital inpatient | Helfer Skin Tap | Standard technique | Simple descriptive pain intensity and Visual Analogue Scale | During injection |
| Hassnein and Soliman [40] 2016 | Quasi-experimental (crossover design) | Egypt | 100 | M38 F62 | 20–60 | Hospital inpatient | Helfer Skin Tap | Standard technique | Universal Pain Assessment Tool | Within 1 min of injection |
| Khanra et al. [22] 2018 | RCT | India | 60 | M29 F31 | 20–60 | Hospital inpatient | Helfer Skin Tap | Standard technique | Numerical Rating Scale | 8AM and 8PM of the 1st + 2nd day following injection |
| Çelik and Korshid [20] 2015 | RCT | Turkey | 180 | M75 F105 | 18–80 | Hospital outpatient | Shotblocker | 1. Standard technique (no Shotblocker) 2. Placebo control (smooth aspect Shotblocker) | Visual Analogue Scale | After injection |
| Emel et al. [35] 2017 | RCT | Turkey | 242 | M42 F200 | 18–31 | Community | ShotBlocker | No ShotBlocker | Visual Analogue Scale | Immediately after injection |
| Kanika and Rani [21] 2011 | Quasi-experimental (crossover design) | India | 30 | M14 F16 | 18–59 | Hospital inpatient | Massage after injection | No massage after injection | Visual Analogue Scale and Verbal Rating Scale | <5 mins 30 mins 1 hour |
| Keen et al. [37] 1986 | Quasi-experimental | USA | 50 | M37 F13 | 21–57 | Hospital inpatient | Z track technique | Standard technique | 4-point Likert Scale | Immediately after injection and 12 hours after |
| Najafidolatabad et al. [27] 2010 | RCT | Iran | 90 | M0 F90 | 18–60 | Hospital inpatient | Prone position and Z track | Prone position and air lock technique | Visual Analogue Scale | Unclear |
| Kara and Güneş [32] 2016 | RCT (crossover design) | Turkey | 86 | M29 F46 (11 lost to follow up) | Mean 46.4 (SD 15.2) | Hospital inpatient | 1.Prone position one foot internally rotated 2. Z track technique† | Prone position toes pointed down | Visual Analogue Scale | Unclear |
| Yilmaz et al. [36] 2016 | RCT | Turkey | 66 | M34 F26 (6 excluded) | 18–65 | Hospital inpatient | Z track technique | Standard technique | Visual Analogue Scale | During injection |

(*Continued*)

**Table 1.** (Continued)

| Study (Year of publication) | Design | Country | Sample Size | Gender | Age Range* (years) | Population | Intervention to reduce pain | Comparator | Pain Measurement Tool | Timing of pain assessment |
|---|---|---|---|---|---|---|---|---|---|---|
| Alavi [41] 2007 | Quasi-experimental (crossover design) | Iran | 64 | M32 F32 | 15–59 | Hospital outpatient | Pressure to acupressure point UB31 | Standard Protocol (WHO guidelines) | Visual Analogue Scale | Not stated |
| Suhrabi and Taghinejad [42] 2014 | Quasi-experimental | Iran | 150 | M0 F150 | 15–55 | Hospital inpatient | Pressure to acupressure point UB32 | Standard technique | Visual Analogue Scale | Not stated |
| Raddadi et al. [19] 2017 | RCT | Iran | 90 | M0 F90 | 16–65 | Emergency Department | 1. Pressure to acupressure point BL32 | Standard Protocol (WHO guidelines) | Visual Analogue Scale | Immediately after injection |
| | | | | | | | 2. Pressure and pinching to acupressure point L14 | | | |
| Najafi et al. [43] 2018 | Quasi-experimental (crossover design) | Iran | 48 | M0 F48 | 17–39 | Hospital inpatient | 1. Pressure to acupressure point UB32 | Standard technique | Visual Analogue Scale | Immediately after injection |
| | | | | | | | 2. Pressure to sham acupressure point | | | |
| Rock [44] 2000 | Quasi-experimental | Australia | 70 | Not stated | Not stated | Hospital outpatient | Two-needle technique | One-needle technique | Visual Analogue Scale | Not stated |
| Ağaç and Güneş 2011 [29] | RCT | Turkey | 100 | M65 F35 | 18–54 | Hospital inpatient | Two-needle technique | One-needle technique | Numerical Rating Scale | Immediately after each injection |
| Ozdemir et al. [45] 2013 | Quasi-experimental (crossover design) | Turkey | 25 | M15 F10 | 18–80 | Hospital inpatient | 30 second injection | 10 second injection | Visual Analogue Scale | Immediately after injection + every 5 mins for the first 35 mins |
| Tuğrul and Khorshid [46] 2014 | Quasi-experimental (crossover design) | Turkey | 60 | M29 F31 | 18–62 | Hospital outpatient | Injection speed 5seconds/mL | Injection speed 10seconds/mL | Visual Analogue Scale | Immediately after injection |
| Bartell et al [47] 2008 | RCT (Factorial design) | USA | 80 | Not available (majority female) | Mean 26.7 (SD 0.99) | Community | Cold needle (-20˚C) | Room temperature needle | Visual Analogue Scale | Immediately after each injection and 2 to 4 days later |
| Thomas et al. [48] 2019 | RCT (crossover design) | India | 100 | Not stated (5 lost to follow up) | 13–45 | Hospital inpatient | Cold needle (0–2˚C) | Room temperature needle | Numerical Rating Scale | During Injection |
| Güneş et al. [31] 2013 | Quasi-experimental (crossover design) | Turkey | 70 | M32 F38 | 51.5 (SD 12.4) | Hospital inpatient | Right ventrogluteal site, patients positioned lateral with uppermost extremities in flexion (injections with airlock technique) | Left dorsogluteal site, patients placed in prone position with extremities internally rotated (injections with airlock technique) | Visual Analogue Scale | Immediately after each injection |

(*Continued*)

**Table 1.** (Continued)

| Study (Year of publication) | Design | Country | Sample Size | Gender | Age Range* (years) | Population | Intervention to reduce pain | Comparator | Pain Measurement Tool | Timing of pain assessment |
|---|---|---|---|---|---|---|---|---|---|---|
| Yilmaz et al. [30] 2016 | RCT | Turkey | 60 | M25 F35 | >18 years | Hospital inpatient | Injection to the ventrogluteal site with or without the airlock technique | Injection to the dorsogluteal site with or without the airlock technique | Visual Analogue Scale | Immediately after each injection |
| Maiden et al. [49] 2003 | RCT | Australia | 150 | M92 F58 | 16–91 | Emergency Department | ǂ 1. Rubbed vaccine (~27˚C) <br> 2. Warmed vaccine (~29˚C) | Cold Vaccine (~19˚C) | McGill Present Pain Intensity Questionnaire | 5 mins 24hrs 48 hrs |
| Farhadi and Esmailzaseh [50] 2011 | RCT | Iran | 60 | M30 F30 | 15–50 | Hospital outpatient | Ice applied to skin prior to injection | Standard technique | Visual Analogue Scale | Immediately after injection |

| Study (Year of publication) | Design | Country | P(population) I (intervention) C (comparator) O (outcome) S (study) | Number of included studies | Study types | Search period | Age | IMI interventions identified | | |
|---|---|---|---|---|---|---|---|---|---|---|
| Şanlialp et al. [15] 2019 | Systematic review & meta-analysis | Turkey | P- Adults (≥ 18 years) administered IM I in any setting I- physical procedural interventions used during IMI C- any physical procedural IMI technique O- pain scale measures S- RCTS and quasi-experimental studies | 15 | 9 RCTs 6 Quasi-experimental studies | Till November 2017 | Adult | **1.** Manual pressure <br> 2. ShotBlocker <br> 3. Z track technique <br> 4. Airlock technique <br> 5. Post injection massage of injection site <br> 6. Injection speed <br> 7. Two-needle technique <br> 8. Acupressure <br> 9. Gluteal injection site | | |

*where age range not available mean and standard deviation (SD) provided

† technique used in addition to comparator.

ǂ- Warmed vaccine- vaccine warmed in a 37˚C warming cupboard for 5 minutes; rubbed vaccine- vaccine rubbed for 1 minute between nurse' hand.

studies applied pressure for 10 seconds prior to giving a vaccination to the deltoid muscle in healthy volunteers [28, 34, 38], and in the fourth study, participants received benzathine penicillin injection into the gluteal muscle [39].

The Helfer skin tap (rhythmic tapping over the skin at the site of injection to relax the muscle before and during the injection) injection technique also reduced injection pain. These studies included one RCT [22] and two quasi-experimental studies [40, 51] involving inpatients who received injections into the gluteal muscle. Post injection massage reduced injection pain compared to the control group, in a quasi-experimental study of hospital inpatients receiving gluteal injections of an analgesic or vitamin K [21].

The acupressure IMI technique involved applying manual pressure for 1 minute to the acupoint UB31/BL32 located in the inner upper quadrant of the dorsogluteal muscle, followed by rhythmic application of pressure with the thumb three times to the acupoints. This was found to be beneficial in reducing injection associated pain in a cohort of hospital inpatients who received antibiotic injections [19, 41, 42] or magnesium sulphate [43].

Najafidolatabad et al [27] compared two different interventions to reduce IMI in a RCT of hospital inpatients receiving analgesic injections at the gluteal site. The Z track technique involved pulling the overlying skin and subcutaneous tissues approximately 2.5 cm laterally,

| Study (author and publication year) | Sequence Generation | Allocation Concealment | Blinding of participants | Blinding of personnel | Blinding of outcome assessors |
|---|---|---|---|---|---|
| **Helfer skin tap** | | | | | |
| Khanra et al. [22] 2018 | + | ? | ? | ? | ? |
| **ShotBlocker** | | | | | |
| Çelik and Korshid [20] 2015 | ? | ? | - | - | ? |
| Emel et al. [35] 2017 | + | ? | - | - | ? |
| **Z track** | | | | | |
| Najafidolatabad et al. [27] 2010 | ? | ? | - | - | - |
| Kara and Güneş [32] 2016 | - | - | ? | - | + |
| Yilmaz et al. [36] 2016 | ? | ? | ? | - | + |
| **Acupressure** | | | | | |
| Raddadi et al. [19] 2017 | + | ? | ? | ? | ? |
| **Two-needle technique** | | | | | |
| Ağaç and Güneş [29] 2011 | ? | ? | + | ? | + |
| **Cold needle** | | | | | |
| Bartell et al. [47] 2008 | ? | - | + | + | ? |
| Thomas et al. [48] 2019 | + | ? | + | - | ? |
| **Gluteal injection site** | | | | | |
| Yilmaz et al. [30] 2016 | ? | ? | - | - | + |
| **Temperature of injectate** | | | | | |
| Maiden et al. [49] 2003 | + | + | + | - | + |
| **Application of ice on injection site prior to injection** | | | | | |
| Farhadi and Esmailzadeh [50] 2011 | ? | ? | - | - | + |

**Fig 2. Risk of bias assessment for included randomised controlled trials.** Red–high risk of bias, yellow–unclear, green–low risk of bias.

holding the skin taut with the non-dominant hand, and then injecting the medication. The second intervention was the air lock technique in which additional air is drawn up into the syringe after the injectate. The air is then injected into the participant along with the injectate. Injection site pain was reported to be significantly lower with the use of the air lock technique compared to the Z track technique.

Application of ice on the injection site for 30 seconds prior to injection significantly reduced injection pain in a RCT of outpatients receiving IM benzathine penicillin [50].

## IMI techniques with reported non-significant reduction in injection pain

The Z track technique (3 studies), altering the temperature of injectate (1 study), and acupressure to L14 acupoint (1 study) did not significantly affect injection pain (Table 3).

**Table 2. Risk of bias for included quasi-experimental studies and systematic review.**

| Study (author and publication year) | Bias due to confounding | Bias in selection of participants | Bias in classification of intervention | Bias due to deviations from intended interventions | Bias due to missing data | Bias in measurement of outcomes | Bias in selection of the reported result | Overall bias |
|---|---|---|---|---|---|---|---|---|
| **Manual pressure** | | | | | | | | |
| Barnhill et al. [34] 1996 | Moderate | Low | Low | Low | Moderate | Moderate | Low | Moderate |
| Chung et al. [38] 2002 | Moderate | Low | Low | Low | Low | Moderate | Low | Moderate |
| Zore and Dias [39] 2014 | Low | NI | Low | Low | Moderate | Moderate | Low | Moderate |
| Öztürk et al. [28] 2016 | Low | Low | Low | Low | Low | Low | Low | Low |
| **Helfer skin tap** | | | | | | | | |
| Shah and Narayan [51] 2016 | Low | Low | Low | Low | Low | Moderate | Low | Moderate |
| Hassnein and Soliman [40] 2016 | Low | Low | Low | Low | Low | Low | Low | Low |
| **Massage after injection** | | | | | | | | |
| Kanika et al. [21] 2011 | Low | Low | Low | Low | Low | Moderate | Low | Moderate |
| **Z track** | | | | | | | | |
| Keen [37] 1986 | Moderate | Low | Low | Low | Low | Low | Low | Moderate |
| **Acupressure** | | | | | | | | |
| Alavi [41] 2007 | Low | Low | low | Low | Low | Low | Low | Low |
| Suhrabi and Taghinejad [42] 2014 | Low | NI | Low | Low | Low | Moderate | Low | Moderate |
| Najafi et al. [43] 2018 | Low | Low | Low | Low | Low | Low | Low | Low |
| **Two-needle technique** | | | | | | | | |
| Rock [44] 2000 | Low | Low | Low | Low | NI | Moderate | Low | Moderate |
| **Injection speed** | | | | | | | | |
| Ozdemir et al. [45] 2013 | Low | Low | Low | Low | Low | Low | Low | Low |
| Tuğrul and Khorshid [35] 2014 | Low | Low | Low | Low | Low | Moderate | Low | Moderate |
| **Gluteal injection site** | | | | | | | | |
| Güneş et al. [31] 2013 | Low | Low | Low | Low | Low | Moderate | Low | Moderate |

| | | | Systematic review | | |
|---|---|---|---|---|---|
| Study (author and publication year) | Bias from study eligibility criteria | Bias in identification and selection of studies | Bias in data collection and appraisal | Bias in synthesis and findings | Overall bias |
| Şanlialp et al. [15] 2019 | High | Low | Low | Low | Low |

NI- No information.

**Table 3. Pain scores following interventions to reduce intramuscular injection pain.**

| Study (Year of publication) | Indication for IM injection | Substance injected | Site of injection | Pain Measurement Tool | Pain descriptors | Mean score intervention | Mean score comparator | Statistical Significance |
|---|---|---|---|---|---|---|---|---|
| **Manual Pressure** | | | | | | | | |
| Barnhill et al. [34] 1996 | Vaccination | Immunoglobulin | Gluteal | Visual Analogue Scale | 100mm "no pain" to "pain as bad as it could be" | 13.6* | 21.5* | p = 0.03 |
| Chung et al. [38] 2002 | Vaccination | Hepatitis A/B vaccine | Deltoid | Verbal rating Scale (Cantonese) | 0 to10 | 1.77 (SD1.49) | 2.86 (SD 1.58) | p<0.0001 |
| Zore and Dias [39] 2014 | Infection | Benzathine penicillin | Not stated | Numerical Rating Scale and behavioural observation | Not stated | Not stated | Not stated | P<0.05 |
| Öztürk et al. [28] 2016 | Vaccination | Hepatitis A/B vaccine | Deltoid | Numerical Rating Scale | 0 to10 | 3.17 (SD 1.95) | 4.0 (SD 2.20) | p = 0.002 |
| **Helfer Skin Tap** | | | | | | | | |
| Shah and Narayanan [13] 2016 | Pain relief (post orthopaedic surgery) | Diclofenac | Ventrogluteal | Visual Analogue Scale | Not stated | Not stated | 1.41 (SD 0.51) ¤ | <0.001 |
| Hassnein and Soliman [40] 2016 | Pain relief | Analgesic or vitamin | Dorsogluteal | Universal Pain Assessment Tool (Verbal description [VDS], Wong Baker face grimace [WFGS] and activity tolerance scales [ATS]) | VDS (0 to 10)-"no pain"(0), to "worst pain"(9 to10) WFGS- "alert smiling" (0) to "eyes closed" (9 to 10) ATS- "no pain" (0), "bed rest required" (9 to 10) | Not stated | Not stated | VDS, p = 0.002; WFGS, p = 0.002; ATS, p = 0.003 |
| Khanra et al. [22] 2018 | Pain relief (post-operative patients) | Analgesic | Dorsogluteal | Numerical Rating Scale | Not stated | 1st day 8 am; 1.97(0.4) | 1st day 8 am; 4.7(0.82) | 1st day 8 am; p<0.001 |
| | | | | | | 1st day 8pm; 0.73 (0.5) | 1st day 8pm; 4.03 (0.79) | 1st day 8pm; p<0.001 |
| | | | | | | 2nd day 8 am; 1.37(0.55) | 2nd day 8 am; 4.90(0.65) | 2nd day 8 am; p<0.001 |
| | | | | | | 2nd day 8 pm 0.40 (0.57) | 2nd day 8 pm 4.30 (0.90) | 2nd day 8 pm; p<0.001 |
| **ShotBlocker** | | | | | | | | |
| Çelik and Khorshid [20] 2015 | Pain relief | Diclofenac | Ventrogluteal | Visual Analogue Scale | 0 "no pain" to 100 "worst pain" | 7.85 (SD 7.03) | 26.7 (SD 20.3)-comparator 20.3 (SD 14.4)-placebo | p<0.001 for intervention vs placebo vs comparator |
| Emel et al. [35] 2017 | Vaccination | Hepatitis B vaccine | Deltoid | Visual Analogue Scale | Not stated | 33.8 (SD 26.1) | 33.0 (SD 23.9) | p = 0.796 |
| **Massage after injection** | | | | | | | | |
| Kanika and Rani [21] 2011 | Pain relief | Diclofenac and/or vitamin K | Gluteal | Visual Analogue Scale (VAS) Verbal rating Scale (VRS) | VAS; 0 to 10 VRS; 0 "no pain" to 5 "worst possible" | VAS (<5mins); 2.02 | VAS (<5mins); 2.9 | VAS at 5mins, 30 mins and 1 hour P<0.05 |
| | | | | | | VAS (30mins); 1.13 | VAS (30mins); 1.82 | VRS at 5mins, 30 mins and 1 hour P<0.05 |
| | | | | | | VAS (1 hour); 0.3 | VAS (1 hour); 0.98 | |
| | | | | | | VRS (<5mins); 2.02 | VRS (<5mins); 2.57 | |
| | | | | | | VRS (30mins); 0.93 | VRS (30mins); 1.43 | |
| | | | | | | VRS (1 hour); 0.13 | VRS (1 hour); 0.62 | |
| **Z track technique** | | | | | | | | |
| Keen et al. [37] 1986 | Pain relief | Meperidine hydrochloride alone or combined with promethazine hydrochloride | Ventrogluteal | 4-point Likert Scale | None, mild, moderate, severe | Not stated | Not stated | The incidence and severity of discomfort were significantly lower in the intervention compared to the comparator group at selected time intervals and not all time intervals. |

*(Continued)*

**Table 3.** (*Continued*)

| Study (Year of publication) | Indication for IM injection | Substance injected | Site of injection | Pain Measurement Tool | Pain descriptors | Mean score intervention | Mean score comparator | Statistical Significance |
|---|---|---|---|---|---|---|---|---|
| Najafidolatabad et al. [27] 2010 | Pain relief | Tramadol | Not stated | Visual Analogue Scale | 0-10cm | 4.56 (SD 1.66) | 2.84 (SD 1.24) | p<0.05 (air lock technique superior to z track) |
| Kara and Güneş [32] 2016 | Pain relief (surgery associated pain) | Diclofenac | Dorsogluteal | Visual Analogue Scale | 0-100mm 0 "no pain" to 100 "extreme pain' | Foot internally rotated, 0.95 (SD 0.91) Z track†, 1.25 (SD 1.44) | Toes pointing down 1.49 (SD 1.28) | p = 0.009 for comparison of foot internally rotated vs Z track† vs toes pointing down |
| Yilmaz et al. [36] 2016 | Pain relief | Diclofenac | Gluteal | Visual Analogue Scale | 0 "no pain to 100 "most intense pain" | 28.3 (SD 23.0) | 36.4 (SD 32.5) | p = 0.336 |
| **Pressure to acupressure point** | | | | | | | | |
| Alavi [41] 2007 | Infection | Penicillin G procaine | Dorsogluteal | Visual Analogue Scale | 0 to 10 | 3 (SD 2) | 5 (SD 2) | p<0.001 |
| Suhrabi and Taghinejad [42] 2014 | Infection | Penicillin | Gluteal | Visual Analogue Scale | 0 "no pain" to 10 "most severe pain" | 2.34 (SD 1.47) | 7.12 (SD 1.88) | P<0.001 |
| Raddadi et al. [19] 2017 | Infection | Procaine Penicillin | Dorsogluteal | Visual Analogue Scale | 0 "no pain" to 10 "severest pain" | BL32 acupoint, 1.76 (SD 2.45) | 2.76 (SD 1.80) | p = 0.006 for comparator vs BL32 |
| | | | | | | L14 acupoint, 2.33 (SD 1.80) ᵀ | | p = 0.051 for comparator vs L14 |
| | | | | | | | | p = 0.030 BL32 vs L14 |
| Najafi et al. [43] 2018 | Pre-eclampsia/ Eclampsia | Magnesium sulphate | Dorsogluteal | Visual Analogue Scale | 0 "no pain" to 10 "worst imaginable pain" | Intervention, 1.94 (SD 1.27) | 7.22 (2.08) | p<0.001 for Intervention vs Comparator; Intervention vs Sham; and Sham vs comparator |
| | | | | | | Sham intervention, 4.75 (SD 2.08) | | |
| **Two-needle technique** | | | | | | | | |
| Rock [44] 2000 | Mental Health | Neuroleptic medication | Dorsogluteal | Visual Analogue Scale | 0 to 100 | Not stated | Not stated | p = 0.895 |
| Aǧaç and Güneş [29] 2011 | Pain relief (secondary to a traffic accident) | Diclofenac | Dorsogluteal | Numerical Rating Scale | 0 "no pain" to 10 "worst imaginable pain" | 5.53 (SD 1.64) | 6.43 (SD 1.35) | p<0.001 |
| **Variable injection speed** | | | | | | | | |
| Ozdemir et al. [45] 2013 | Dermatological condition | Methylprednisolone | Dorsogluteal | Visual Analogue Scale | 0 "no pain" to 10 "unbeareable pain" | 3.4 (SD 1.6);0 min | 4.1 (SD 2.1); 0 min | p = 0.016; 0 min |
| | | | | | | 3.1 (SD 1.5); 5 min | 3.6 (SD 1.9); 5 min | p = 0.136; 5 min |
| | | | | | | 2.2 (SD 1.4); 10 min | 3.0 (SD 1.7); 10 min | p = 0.036;10 min |
| | | | | | | 1.2 (SD 1.2); 15min | 2.2 (SD 1.6); 15min | p = 0.004; 15min |
| | | | | | | 0.5 (SD 0.8); 20 min | 1.5 (SD 1.5); 20 min | p = 0.008;20 min |
| | | | | | | 0.1 (SD 0.4); 25 min | 0.7 (SD 1.1); 25 min | p = 0.013; 25 min |
| | | | | | | 0.1 (SD 0.4); 30 min | 0.2 (SD 0.5); 30 min | p = 0.265; 30 min |
| | | | | | | 0.0 (SD 0.0); 35 min | 0.1 (SD 0.2); 35 min | p = 0.327; 35 min |
| Tuǧrul and Khorshid [46] 2014 | Infection | Penicillin G | Ventro/ dorsogluteal | Visual Analogue Scale | 0 "no pain" to 10 "excruciating pain" | 43.8 (SD 22.2); VG 45.8 (SD 20.3); DG | 48.6 (SD 24.5); VG 49.6 (24.5); DG | p = 0.230 for; VG |
| | | | | | | | | p = 0.067 for DG |
| **Cold needle technique** | | | | | | | | |
| Bartell et al. [47] 2008 | Vaccination | influenza vaccination saline | Deltoid | Visual Analogue Scale | 10 cm "no pain" to "most painful injection ever." | 32.2 (SE 3.2); Flu vaccine 25.2 (SE 3.0); Saline | 36 (SE 3.8); Flu vaccine 23.7 23.7(SE 3.2); Saline | p = 0.450 for cold needle vs room temperature needle with Flu vaccine |
| | | | | | | | | p = 0.733 for cold needle vs room temperature needle with Saline |
| Thomas et al. [48] 2019 | Infection | Penicillin benzathine | Not stated | Numerical Rating Scale | 0 "no pain" to 10 "worst imaginable pain" | 3.37 (SD 1.75) | 5.58 (1.68) | P = 0.001 |
| **Gluteal injection site (VG vs DG)** | | | | | | | | |

(*Continued*)

**Table 3.** (Continued)

| Study (Year of publication) | Indication for IM injection | Substance injected | Site of injection | Pain Measurement Tool | Pain descriptors | Mean score intervention | Mean score comparator | Statistical Significance |
|---|---|---|---|---|---|---|---|---|
| Güneş et al. [31] | Pain relief (patients with spinal disc herniation) | Diclofenac | Ventro/ dorsogluteal | Visual Analogue Scale | 0 "no pain" to 10 "worst imaginable pain" | 1.24 (1.18) | 1.89 (1.49) | p = 0.019 |
| Yilmaz et al. [30] 2016 | Pain relief | Diclofenac | Gluteal | Visual Analogue Scale | 0 "no pain" to 10 "worst imaginable pain" | 2.53 (SD 2.52); air lock-VG | 3.30 (SD 2.70); air lock-DG | p = 0.197; VG vs DG |
| | | | | | | 2.99 (SD 2.86); without airlock-VG | 3.16 (SD 2.74); without airlock-DG | p = 0.519; VG vs DG |
| **Other Techniques** | | | | | | | | |
| Maiden et al. [49] 2003 | Vaccination | Diphtheria/tetanus | Deltoid | McGill Present Pain Intensity Questionnaire | 0 "no pain" to 5 "excruciating pain" | Median 1.0 (IQR 1.0–2.0)-Warmed vaccine | Median 1.0 (IQR 1.0–2.0)-cold vaccine | p = 0.630 for warmed vs rubbed vs cold vaccine |
| | | | | | | Median 1.0 (IQR 1.0–2.0)-Rubbed vaccine ⱡ | | |
| Farhadi and Esmailzaseh [50] 2011 | Infection | Benzathine penicillin | Dorsogluteal | Visual Analogue Scale | 0 to 10 | 4.47 (1.42) | 7.39 (1.55) | p< 0.001 |

**Systematic review and meta-analysis**

| Study (Year of publication) | Risk of bias of reviewed studies | Inter-study heterogeneity | Pooled outcome | Author's conclusions on IMI techniques and injection pain |
|---|---|---|---|---|
| Şanlialp et al. [15] 2019 | Allocation concealment was unclear and risk from blinding of participant and personnel was unclear or high in RCTs. For quasi-experimental studies, half of the studies reliably measured the outcome. | Significant inter-study heterogeneity | standardised mean difference in pain scores (95%CI) for all intervention techniques vs comparator 0.595(0.417 to 0.773; p = 0.001).<br>Subgroup analysis by IM intervention vs comparator<br>SMD manual pressure, 3 studies 0.557 (0.372 to 0.741, p = 0.001)<br>SMD Z track, 3 studies 0.587 (0.044 to 1.130, p = 0.001)<br>SMD ShotBlocker 1.021 (0.468 to 1.574, p = 0.001)<br>SMD Two needle technique 0.744 (0.335 to 1.154, p = 0.001)<br>SMD Acupressure 0.403(-0.123 to 0.929, p = 0.133)<br>SMD Airlock technique 0.295 (-0.391 to 0.981, p = 0.400)<br>SMD Injection speed 0.352 (0.073 to 0.777, p = 0.105)<br>Subgroup analysis by injection site<br>SMD deltoid, 2 studies 0.545 (0.032 to 1.059, p = 0.037)<br>SMD dorsogluteal, 7 studies 0.493 (0.208 to 0.778, p = 0.001)<br>SMD ventrogluteal 3 studies 0.791 (0.355 to 1.227, p<0.001)<br>SMD DG&VG, 3 studies 0.701 (0.243 to 1.158; p = 0.003) | No evidence for a single IMI technique, however, the VG site, Z track technique, and manual pressure were most effective in reducing IMI pain. Other effective methods were the two-needle technique, post-injection massage, and the ShotBlocker. |

*-Adjusted mean.

¤- Mean difference and standard deviation of mean difference comparator vs intervention.

†- Technique used in addition to comparator.

Ŧ- Two interventions; acupressure to acupoint BL32 or acupoint L14.

VG-Ventrogluteal; DG-dorsogluteal.

SE—Standard error; IQR—Interquartile range.

ⱡ- Warmed vaccine- vaccine warmed in a 37˚C warming cupboard for 5 minutes; rubbed vaccine- vaccine rubbed for 1 minute between nurse' handswp.

Yimaz et al. [36] found that the Z track technique did not confer a significant advantage over the standard IMI technique in reducing injection pain, although Z track reduced drug leakage. Kara and Güneş [32] compared three different techniques with patients in the prone position; (1) 'toes pointing down' only (2) 'internally rotated foot' only and (3) 'toes pointing down' combined with the Z track technique. Injection pain was reduced in the 'toes pointing down' combined with Z track group compared to the 'toes pointing down' only group (mean VAS 1.25 vs 1.49 respectively). However, injections with the 'toes pointing down' combined with Z track were more painful when compared to injections with 'internally rotated feet 'only

(mean VAS 1.25 vs 0.95). Keen et al. [37], also investigated the effect of Z track technique on injection pain in participants and measured the intensity of discomfort (burning, stinging, aching, being sore or hurting when touched or moving the leg) with a 4-point Likert scale. The severity of discomfort (immediately after injection and 3 to 5 hours after initial injection) was similar for both the Z track technique and the standard technique arms, although the Z track was effective at reducing the incidence of selected descriptors of discomfort at other selected time intervals. All three studies involved hospital inpatients receiving IM analgesic injections.

Maiden et al. [49] investigated the influence of the injectate temperature on IMI pain using combined adult diphtheria and tetanus vaccine. "Cold" vaccine with no deliberate warming (mean temperature 19.1˚C), "rubbed" vaccine which was rubbed for 1 minute between the nurses' hands (mean temperature 26.9˚C), and "warmed" vaccine which was placed in a warming cupboard for 5 minutes (mean temperature 28.9˚C) were injected into the deltoid muscle. Across all time points measured, post injection pain was similar regardless of the temperature of vaccine injected (Table 3).

Raddadi et al. [19] investigated the effect of applying acupressure to acupoint L14 (a region between the thumb and index finger) on injection pain. Acupressure to the L14 acupoint did not significantly reduce injection pain when compared to a control group who did not receive acupressure (mean VAS 2.33 vs 2.76), and injection pain was significantly higher in those randomised to the L14 acupoint compared to the BL32 acupoint (local to the gluteal injection site)—mean VAS 2.33 vs 1.76.

## IMI techniques with inconsistent effects on injection pain

The evidence supporting the benefits of IMI techniques such as 'ShotBlocker' (2 studies), two-needle technique (2 studies), injection speed (2 studies), altering the temperature of the injecting needle (2 studies) and using the ventrogluteal or dorsogluteal sites for injection (2 studies), were inconsistent across studies (Table 3).

Çelik and Khorshid [20] reported a significant reduction in post injection pain with the use of the 'ShotBlocker' device (a plastic device used to apply local pressure to the skin) as an intervention in hospital inpatients who received an analgesic injection into the gluteal muscle (mean VAS 7.85 vs 20.3 vs 26.7, intervention vs placebo control vs non-placebo control respectively). However, a larger study (n = 242) by Emel et al. [35] did not find post injection pain to be reduced with use of a 'ShotBlocker' (mean VAS 33.8 vs 33.0, 'ShotBlocker' vs 'No Shot-Blocker'), although in this study patients received a smaller volume injection to the deltoid muscle. Both Çelik's and Emel's studies were RCTs.

Ağaç and Güneş [29] found that a two-needle technique produced less pain compared to a one needle technique (mean VAS 5.53 vs 6.43) in an RCT involving a cohort of road traffic accident trauma patients. Those receiving the two-needle technique had the needle changed after drawing up the injectate and prior to the injection of the analgesic. In contrast, Rock [44] did not find a significant (p = 0.895) reduction in injection site pain when the two-needle technique was compared to a one-needle technique in quasi-experimental study of psychiatric outpatients receiving IMI of neuroleptics. Ağaç and Güneş [29] and Rock [44] used the airlock and Z track injection technique respectively in both the intervention and control groups.

Slow injection (30 seconds) was found to be beneficial in reducing injection pain following injection of 1ml methylprednisolone at the gluteal site compared to a fast injection (10 seconds) [45]. In contrast, Tuğrul and Khorshid [46] found that injection pain following IM injection of 800,000IU penicillin diluted with 2ml of sterile injectable water at a speed of 1ml/ 5s (fast injection) was similar to that of an injection speed of 1ml/10s regardless of whether the injection site was dorsogluteal or ventrogluteal.

In a RCT with factorial design study, Bartell et al. [47] randomised patients to receive influenza vaccination with either a cold (-20°C) or room temperature needle. The same participants were then randomised a second time to receive a saline injection with either a cold or room temperature needle. The mean pain score did not differ significantly between the two groups, regardless of the substance injected. However, in another RCT, IMI with a cold needle (-2 to 0°C) was associated with significantly reduced injection pain compared to injections with room temperature needles. This was a study with hospital outpatients with rheumatic heart disease receiving IMI of benzathine penicillin [48].

Participants in Yilmaz et al. [30] were randomised to receive analgesic injections into either dorsogluteal or ventrogluteal site, injections to the randomised injection sites being administered with and without the use of the airlock injection technique. The reported pain was similar at both injection sites regardless of the use of the airlock injection technique. This finding is in contrast with an earlier study by Güneş et al. [31] where injections to the right ventrogluteal muscle were associated with lower IMI pain compared to the left dorsogluteal muscle in patients receiving analgesic injections with the airlock injection technique.

## Meta-analyses of studies

Meta-analyses of the studies using local pressure (manual pressure, Helfer skin tap, Shot-Blocker, and post injection massage), acupressure, Z track IMI techniques, cold needle technique and the choice of gluteal muscle injection site were performed (Figs 3–7). Application of local pressure to the injection site (10 studies, SMD = -1.44 [95% CI -1.99,-0.89]) or to specific acupressure points—acupressure (4 studies, SMD = -1.62 [95% CI -2.80,-0.44]) were effective in reducing injection site pain (Figs 3 and 5). Although the direction of effect for most of the studies in these meta-analyses was consistent, there was considerable heterogeneity between the studies ($I^2$ = 95% [92,97] and 96%[92,98] for local pressure to injection site and acupressure respectively). For gluteal injections administered with the airlock technique, use of the ventrogluteal injection site conferred some benefit on injection pain (2 studies, SMD = -0.43 [-0.81,-0.06]), and between study heterogeneity was low ($I^2$ = 0%)—Fig 7. Both the Z track and cold needle IMI techniques reduced injection pain but did not reach statistical significance. For the Z track technique (2 studies, SMD = -0.20 [95% CI -0.41,0.01]), heterogeneity was low between studies ($I^2$ = 0%)—Fig 4. However, for the cold needle technique (2 studies, SMD = -0.73 [95%CI -1.83,0.37]) heterogeneity was high between studies ($I^2$ = 92% [72,98])—Fig 6.

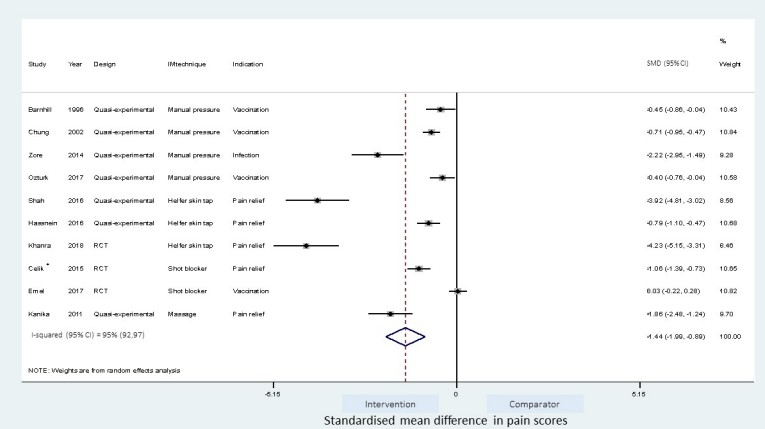

**Fig 3. Effect on pain of techniques applying local pressure to the IMI site.** *-Placebo control and non-placebo control arms combined.

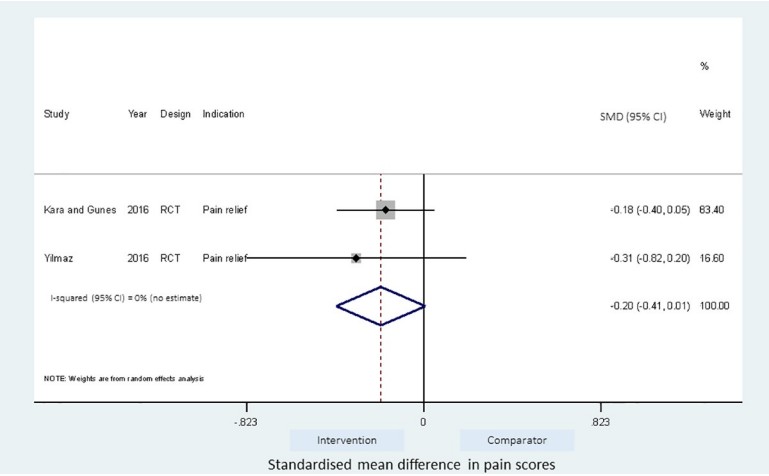

**Fig 4. Effect of Z track IMI technique on injection pain.**

**Studies investigating the application of local pressure.** Sub-group analyses to explore heterogeneity, and identify the effects of closely related IMI techniques and the type of study design on the pooled estimate were performed for studies investigating the application of local pressure.

**Fig 5. Effect of acupressure IMI technique on injection pain.** *Combined estimates from acupressure to BL32 and L14 pressure points.

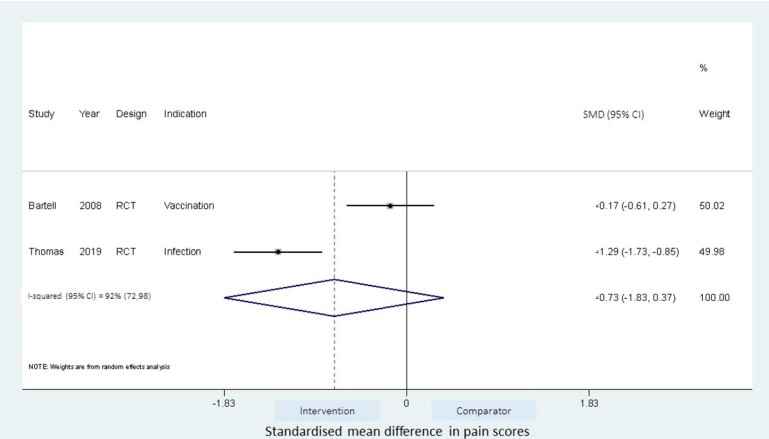

**Fig 6. Effect of cold needle IMI technique on injection pain.**

A meta-regression on the differences between the type of pressure application and the effect size was not feasible owing to the relatively large number (four) of different techniques. However, as shown in Table 4, the pooled estimate for the ShotBlocker technique failed to reach statistical significance (2 studies, SMD = -0.51 [-1.58,0.56]). Application of manual pressure (4 studies; SMD = -0.85[-1.36,-0.33]) and Helfer skin tap (3 studies, SMD = -2.95 [-5.51,-0.39]) to the injection site were effective in reducing pain. Post-injection site massage reduced pain (SMD = -1.86 [-2.48,-1.24]) but this estimate was based on a single study.

The study design (RCT vs quasi-experimental) did not affect the effect size (p = 0.842) when explored by a meta-regression (S1 Table), and the pooled estimate for RCTs (3 studies) and quasi-experimental studies (7 studies) were similar (SMD for RCTs = -1.67[-3.19,-0.14]; SMD for quasi-experimental studies = -1.36[-1.94, -0.78])—Table 4.

Asymmetry was observed in the funnel plot of the 10 studies investigating the application of local pressure to the injection site (Fig 8A). The asymmetry was further examined with a contour enhanced funnel plot and Egger's meta-regression test for small study effect. Interventions from small studies were highly statistically significant (Fig 8B) and Egger's test for small study bias was also statistically significant (coefficient of -9.09 [95%CI, -13.4, -4.78], p = 0.001).

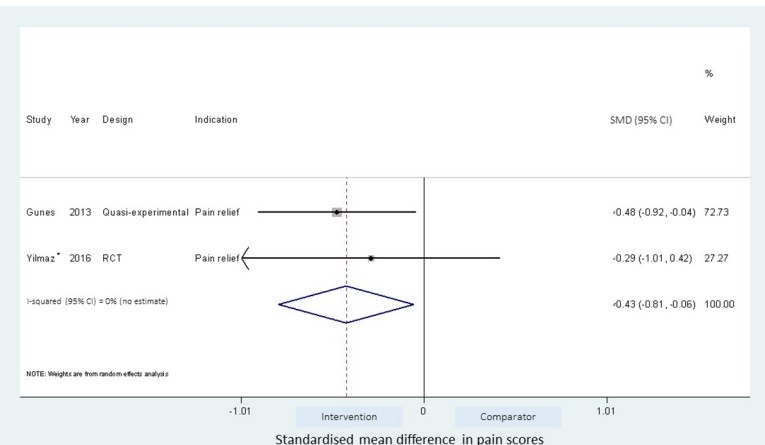

**Fig 7. Effect of gluteal injection site on injection pain.** Intervention is ventrogluteal (VG) injection site and comparator is the dorsogluteal (DG) injection site. *- results of injection to either VG or DG site with airlock IMI technique.

**Table 4. Sub-group analyses of IMI techniques applying pressure to the IMI site.**

| Meta-analysis | Number of studies | Pooled SMD(95%CI) | P value | Heterogeneity $I^2$ (95%CI) |
|---|---|---|---|---|
| All pressure based studies | 10 | -1.44 (-1.99,-0.89) | >0.001 | 95% (92, 97) |
| Pressure techniques | | | | |
| Manual pressure | 4 | -0.85 (-1.36,-0.33) | 0.001 | 86% (65, 94) |
| Helfer skin Tap | 3 | -2.95 (-5.51,-0.39) | 0.024 | 98% (95, 99) |
| ShotBlocker | 2 | -0.51 (-1.58,0.56) | 0.347 | 96% (no estimate) |
| Post injection massage | 1 | -1.86 (-2.48,-1.24) | >0.001 | NA |
| Study design | | | | |
| RCTs | 3 | -1.67 (-3.19,-0.14) | 0.032 | 98% (96, 99) |
| Quasi experimental | 7 | -1.36 (-1.94,-0.78) | >0.001 | 93% (87, 96) |

NA—not assessed.

In S2 Table, sensitivity analyses were reported and the estimate of the intervention effect using a fixed effects meta-analysis for IMI techniques employing local pressure application differed considerably from the random effects estimate (fixed vs random effects SMD = - 0.74 [-0.86,-0.63 vs 1.44[-1.99,-0.89]), reinforcing the likely bias from small studies in the meta-analysis. Because of this, four small studies [21, 22, 34, 39] (parallel studies with study size ≤ 100 and cross over studies with studies size ≤ 50) were excluded from a further meta-analysis of local pressure techniques resulting in the effect size estimate being reduced by 44%

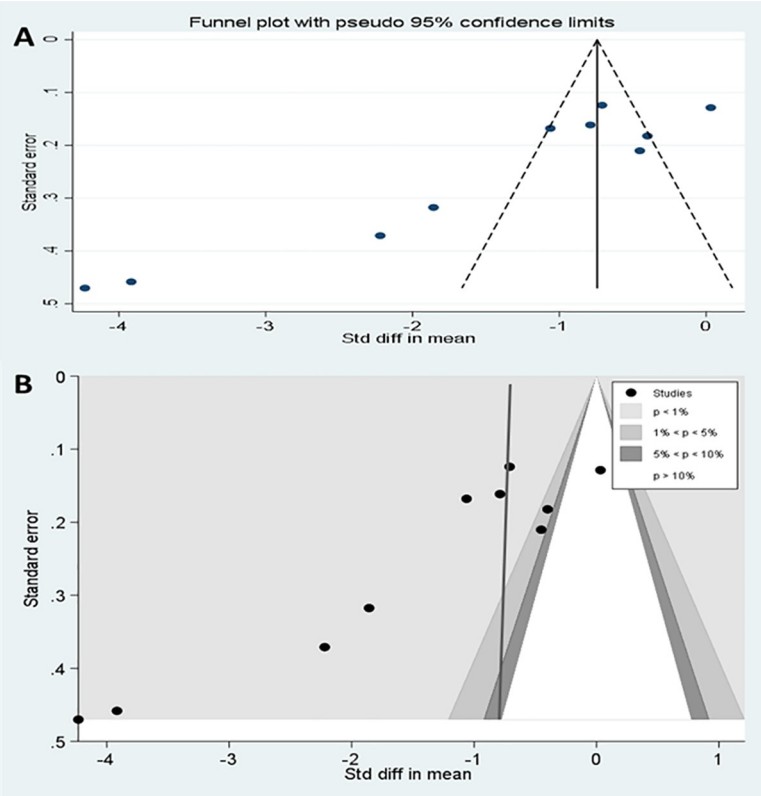

**Fig 8. Publication bias of meta-analysis of interventions applying pressure to injection site.** (A) Funnel plots of studies (B) Contour enhanced funnel plot.

(All pressure studies vs large pressure studies only SMD = 1.44[-1.99,-0.89] vs—1.00[-1.56,-0.44]).

**Studies investigating other IMI techniques.** The Z track, acupressure, cold needle IMI technique and gluteal site injection meta-analyses lacked sufficient power to explore subgroup differences and biases. Only sensitivity analyses on the use of random effect and correlation in cross-over trials were explored.

The fixed and random effects estimates for the Z track were similar. The pooled estimates for the Z track technique were also similar regardless of the cross over design of included studies. Najafidolatbad et al. [27] was excluded from the meta-analysis of the Z track IMI intervention because the study compared two different techniques with no control arm. The inclusion of the Najafidolatbad et al. [27] study reversed the direction of the pooled estimate for the Z track IMI technique, indicating that the technique caused more pain than the comparator (with vs without Najafidolatbad et al. [27]; SMD = 0.22[-0.64,1.09] vs -0.20[-0.41,0.01]). The between study heterogeneity also increased by 93% points (S3 Table).

The fixed and random effects estimates, and the pooled estimates incorporating or ignoring the crossover design were similar in the meta-analysis of the acupressure technique (S4 Table).

For the cold needle technique, the estimate from the fixed effect analysis was significant (fixed vs random effects; SMD = -0.73[-1.04,-0.42] vs—0.73[-1.83,0.37]). This difference in the CIs around the estimate is associated with the smaller calculated standard error in the fixed effect model compared to the random effect model, however given the large heterogeneity between studies ($I^2$ = 92%[72,98]), the fixed effect meta-analyses may be an overestimation of the true effect. The pooled estimates were however similar regardless of the cross over design of included studies (S5 Table).

The fixed and random effects estimates for the gluteal injection site meta-analyses (ventrogluteal vs dorsogluteal) were similar (S6 Table). However, the 95%CI around the estimate for the acupressure and gluteal injection site meta-analyses were narrower when the cross-over design was ignored.

## Discussion

Overall, we identified a variety of intramuscular injection (IMI) techniques which reduce injection site pain amongst diverse patient groups and healthcare settings. We found some evidence supporting the use of manual pressure at the site of injection (4 studies, SMD = -0.85[-1.36,-0.33]) and the Helfer skin tap (3 studies, SMD = -2.96 [-5.51,-0.39]). Manual pressure involves sustained application of pressure for 10 seconds with the thumb/fingers [34] while the Helfer skin tap generally involves short repeated pressure application (tapping) with the thumb/fingers for several seconds to relax the muscle prior to injection [22, 51, 52], this may either be preceded [40] or followed by [51, 52] making a large 'V' with the thumb and index finger [51] before inserting the injection needle. However, the evidence supporting these techniques was mostly from small non-randomised studies, and pooled estimates were characterised by significant unexplained heterogeneity which may reflect variations in the patient population and/or study design.

There were contradictory findings for the efficacy of the ShotBlocker technique—another local pressure type technique—on injection pain. Emel et al. [35] found injection associated pain to be similar in both the ShotBlocker and control arm, in contrast to a beneficial effect reported by Çelik and Khorshid [20, 35]. It is possible that the inconsistency in findings relates to the larger study population, smaller injection volume and healthier patient group in the Emel et al. [35] study. Post injection massage was found to be beneficial in reducing injection pain in a single unblinded, quasi-experimental study of 30 participants [21] but further larger studies are required to confirm any benefit.

A meta-analysis combining the ten studies with interventions based on applying local pressure suggests that this approach may be beneficial in reducing injection pain (10 studies, SMD -1.44 [-1.99,-0.89]). However, the estimate was associated with substantial inter study heterogeneity ($I^2$ = 95% [92,97]) and there is a risk of overestimating the effect size given the inclusion of multiple studies with a small sample size. The effectiveness of applying local pressure on the injection site is supported by the findings of a systematic review in children where stroking the skin or applying pressure close to the injection site reduced IM vaccination pain [14]. Mechanistically the benefits of applying local pressure to the injection site may be explained by the gate-control theory of pain which hypothesises that the stimulation of Aβ afferent nerve fibres (mechanoreceptors) inhibit transmission of nociceptive input (pain) to second-order neurons through gating at the substantia gelatinosa in the dorsal root ganglion of the spinal cord [53, 54]. Our analysis of local pressure based IMI techniques on injection site pain updates and expands that of a previous systematic review in adults [15], and suggests that larger RCTs are required to strengthen the evidence for its use in routine practice. The application of local pressure over the injection site is potentially easy to teach and implement, however, standardisation of the amount of pressure and duration of applied pressure is required.

The Z track technique has been widely recommended to reduce injection pain and drug leakage [2]. However, we found no evidence of the benefit of Z track on injection pain in the pooled estimates of 2 RCTs with low inter-study heterogeneity. Kara and Güneş [32] found a reduction in injection pain when the Z track technique was combined with positioning the patients in a prone position with 'toes pointing down' compared to the patient group in a prone position with 'toes pointing down' alone. However, this failed to reach statistical significance in our meta-analysis and the combined Z track technique with 'toes pointing down' was associated with more injection pain when compared to positioning the patient prone with an 'internally rotated foot' alone. Yilmaz et al. [36] also observed a reduction in IMI pain scores with the use of the Z track technique compared to standard IM technique, but again this failed to reach statistical significance in the primary analysis. There have been reported variations to performing the Z track technique in practice [55] and this may affect the observed effectiveness in different studies. It is possible that recommendations to use the Z track technique may have arisen from evidence gathered from descriptive studies and personal viewpoints [2]

Najafidolatbad et al. [27] found the airlock technique to be significantly more beneficial in reducing injection pain compared to the Z track IMI technique. The airlock technique is based on a similar principle to the Z track, where preventing drug leakage might reduce the risk of pain on injection. At present the benefits of the airlock method remain limited and the addition of an air bubble into the syringe may make delivering the correct volume of drug more difficult [56].

Some recommendations have been made in support of the ventrogluteal site over the dorsogluteal site with respect to injection pain and injection complications [12]. However, we found conflicting evidence for the use of the ventrogluteal site as an intervention to reduce injection pain. Yilmaz et al. [30] investigated the influence of using the dorso or ventrogluteal site on injection pain when combined with the airlock technique, and found a non-significant reduction of pain scores in those randomised to the ventrogluteal site compared to the dorsogluteal site. This trend was also observed when injections were administered without the airlock technique. Güneş et al. [31] found that injections administered with the airlock techniques were significantly less painful when given at the ventrogluteal site compared to the dorsogluteal site. Combining these studies in a meta-analysis suggest that the ventrogluteal site may be a better injection site when considering injection pain (2 studies, SMD, -0.43 [-0.81; -0.06]), but more studies are required to support this possibility.

The use of the acupressure technique was found to significantly reduce injection pain, especially when the acupressure point (UB31/UB32/BL32) was close to the injection site [41–43], in which case the acupressure procedure becomes similar to the other manual pressure interventions–local injection site pressure and the Helfer skin tapping technique. The acupressure technique involves sustained application of pressure to the acupoint for 1 minute followed by short repeated application of pressure two to three times prior to injection [19, 41]. When the application of pressure was to an acupressure point site not local to the injection site, it did not appear to be effective [19]. The pooled estimate from the meta-analysis of acupressure was associated with significant heterogeneity, possibly reflecting variation in how the technique is delivered, study design and/or patient groups.

There was contradictory evidence on the benefits of the two-needle technique over the one needle technique. Ağaç and Güneş [29] found the two-needle technique to be beneficial in reducing injection site pain, but the use of the two-needle or one-needle technique was combined with an additional intervention, the airlock technique. Rock [44] combined either two-needle or one-needle technique with Z track and did not find any significant benefit with the two-needle technique. However, since the airlock technique was found to reduce injection pain when compared to the Z track technique in a separate RCT [27], and there were differences in study design and patient groups of the Ağaç and Güneş [29] and Rock [44] studies (Table 1), the benefit of changing the needle prior injection remains unclear.

The evidence for using a fast or slow injection speed is also unclear, owing to the differences in the design of the two relevant studies. Tuğrul and Korshid [46] found that injection pain following IM injection of procaine penicillin diluted with 2ml of sterile water at speeds of 1ml/5s and 1ml/10s was similar, regardless of whether the injection site was dorsogluteal or ventrogluteal. However, Ozdemir et al. [45] found that a methylprednisolone injection diluted with 1ml sterile water injected over 10seconds was significantly more painful than administering the same injection over 30 seconds at the gluteal site. Both studies were small crossover quasi-experimental studies.

Using a cold needle for the injection procedure reduced injection pain compared to the use of room temperature needle, but this was not found to be statistically significant when the data from studies were pooled (Fig 6). Bartell et al. [47] found no significant benefit on injection pain when the cold needle (-20˚C) was used in administering vaccines to the deltoid muscle in healthy volunteers and this was also the case when saline was administered, although saline injections were generally less painful compared to vaccines. In contrast, Thomas et al. [48] found the cold needle (0–2˚C) technique to be significantly beneficial in reducing injection pain following benzathine penicillin injection. It is possible that the variation in temperature of the needles, the injection site or difference in substance injected may influence the benefit of the cold needle technique. More studies are warranted to confirm the benefits of the cold needle technique.

Other studies explored IMI techniques related to modifying the temperature of injectate [49] or injection site [50]. Warming the injectate to ~27 to 29˚C did not alter the severity of injection pain [49]. However, Farhadi and Esmailzadeh [50] found that cooling the injection site with ice for 30 seconds prior to injection, significantly reduced pain. This may relate to the potential benefit of ice-therapy on pain following soft tissue (including muscular) injury [57] but, further studies are required to substantiate this.

Our systematic review and meta-analysis provides updates on the evidence on intramuscular injection techniques beyond the systematic review of Şanlialp et al. [15]. We identified additional evidence on the manual pressure, Z track, ShotBlocker, and acupressure techniques and report on other IMI techniques including the Helfer Skin tap, application of ice to injection site, and altering the temperature of the injectate. Şanlialp et al. [15] found the

ventrogluteal site, Z track technique, and manual pressure to be the most effective IMI techniques in reducing injection pain, but also found that the two-needle technique, post injection massage and ShotBlocker IMI techniques were beneficial in reducing injection pain. The benefits of the manual pressure technique and post injection massage are similar to our assessment, however, our findings on ShotBlocker, acupressure, two-needle technique and Z track are in contrast to the findings of Şanlialp et al. [15]. We identified more studies on the acupressure and ShotBlocker IMI techniques which have influenced our assessments of these techniques. However, both our study and Şanlialp et al. [15] assessed the same evidence on the two-needle and the Z track technique. In Şanlialp et al. [15], the data from the Rock [44] study was transformed, and the two-needle technique was reported to be significantly beneficial in reducing IMI pain, in contrast to the primary findings. Similarly the data from Najafidolatabad et al. [27] was transformed in Şanlialp et al. [15] and the Z track intervention reported to be significantly beneficial in reducing injection pain compared to the air-lock method, again conflicting with the report of the original study. It is likely that the difference in the number of included studies, methodological differences in combining studies for meta-analysis and interpretation of the direction of effect between our study and that of Şanlialp et al. [15] have account for the difference in our inferences.

A risk of bias assessment found poor reporting of essential design elements including blinding and randomisation in most of the identified RCTs. The risk of bias for the non-randomised studies was moderate in the majority of studies. None of the included experimental studies carried out baseline assessment of pain prior to IMI or commented on how patients receiving an IMI for pain relief might interpret injection pain differently to those who were healthy volunteers and presumed to be pain free. The outcome measures, although similar, differed in their precise wording and how they were delivered across studies. This lack of consistency is potentially important when making comparisons between studies particularly for a subjective outcome measure such as pain. We also identified marked variation in standard IM injection practice—for example, the standard technique in Ozturk et al. [28] involved a two needle technique, for Kara and Güneş [32] it included a two-needle technique, airlock technique, aspiration and injection speed of lml /10sec, and Yilmaz et al. [36] included the airlock technique. This may reflect a non-standard approach in how clinicians are initially taught or subsequently learn about IMI technique.

The review has a number of limitations. First, the search terms chosen were based on scoping searches and prior knowledge of the literature on IMI. The use of additional search terms may have increased the number of records returned, but would have made the number of records requiring review unfeasible and was considered unlikely to identify other highly relevant studies. Searches were limited to the main bibliographic databases and citation searching, without review of potential grey literature. The meta-analyses combined data across studies in order to estimate effect size with more precision than is possible in a single study. The patient population, IMI techniques and outcome measures were not identical across studies and therefore caution is required in the meta-analysis interpretation. Publication bias might account for some of the effect we observed. Smaller trials are, in general, analysed with less methodological rigor than larger studies [58], and an asymmetrical funnel plot suggests that selective reporting may have led to an overestimation of effect size in small trials. However, our assessment is restricted to 10 studies, and publication bias assessment can be limited when evaluating a small number of studies [59]. We also performed a meta-regression on the influence of the experimental design of the studies on local pressure IMI techniques (10 studies), and it is possible that we did not have sufficient power to detect an association between study design and effect size, however, the sub-group analysis shows that the estimates from the combined RCT or quasi-experimental studies were similar in both direction and level of significance (Table 4).

Our review would support future research to further explore whether local pressure based IMI techniques, in combination or individually, can reduce IMI pain. The existing evidence is based on small sample sizes (for instance, n = 340 [manual pressure] and n = 242, [Helfer skin tap]) with a significant risk of bias, and larger randomised controlled trials are therefore warranted.

In conclusion, the current evidence suggests that the application of local pressure, especially manual pressure and/or rhythmic tapping of the injection site prior to injection may be beneficial in reducing IMI pain, but any conclusions are limited by small study sizes, non-standardised interventions, imprecise control groups and wide inter-study heterogeneity.

## Supporting information

**S1 Checklist. PRISMA 2009 checklist.**
(DOC)

**S1 Table. Meta-regression of RCT (n = 3) and quasi-experimental studies (n = 7) on the intervention effect of IMI techniques applying pressure to the IMI site.** *The coefficient estimates how the intervention effect (SMD) differs (RCT studies vs Quasi-experimental studies).
(DOCX)

**S2 Table. Sensitivity analyses: Local pressure IMI techniques.** Ŧ- excludes Barnhill et al. [34], Zore and Dias [39], Khanra et al. [22] and Rani and Prasad [21].
(DOCX)

**S3 Table. Sensitivity analyses: Z track IMI technique.** Ŧ–includes Najafidolatbad et al. [27] study (Z track vs air lock technique).
(DOCX)

**S4 Table. Sensitivity analyses: Acupressure IMI technique.**
(DOCX)

**S5 Table. Sensitivity analyses: Cold needle IMI technique.**
(DOCX)

**S6 Table. Sensitivity analyses: Choice of gluteal IMI site.**
(DOCX)

**S7 Table. Search terms in abstract and title.** *Asterisk used to capture multiple word endings (e.g. technique, techniques).
(DOCX)

## Author Contributions

**Conceptualization:** Oluseyi Ayinde, Rachel S. Hayward, Jonathan D. C. Ross.

**Data curation:** Oluseyi Ayinde, Rachel S. Hayward.

**Formal analysis:** Oluseyi Ayinde.

**Investigation:** Oluseyi Ayinde, Rachel S. Hayward.

**Methodology:** Oluseyi Ayinde, Rachel S. Hayward, Jonathan D. C. Ross.

**Writing – original draft:** Oluseyi Ayinde, Rachel S. Hayward, Jonathan D. C. Ross.

**Writing – review & editing:** Oluseyi Ayinde, Rachel S. Hayward, Jonathan D. C. Ross.

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
