## [Decision Letter · Decision Letter 0]

25 Mar 2021

PONE-D-20-24754

The effect of intramuscular injection technique on injection associated pain; a systematic review and meta-analysis

PLOS ONE

Dear Dr. Ayinde,

Thank you for submitting your manuscript to PLOS ONE. After careful consideration, we feel that it has merit but does not fully meet PLOS ONE’s publication criteria as it currently stands. Therefore, we invite you to submit a revised version of the manuscript that addresses the points raised during the review process.

We look forward to receiving your revised manuscript.

Kind regards,

Mª Ángeles Peña Fernández

Academic Editor

PLOS ONE

Additional Editor Comments (if provided):

Reviewers have done an excellent job of evaluating this paper, authors should include all changes and suggestions that propose them.

Journal Requirements:

2) Thank you for stating the following in the Competing Interests section:

[I have read the journal's policy and the authors of this manuscript have the following

competing interests: JR reports personal fees from GSK Pharma, Hologic Diagnostics,

Mycovia and Janssen Pharma as well as ownership of shares in GSK Pharma and

AstraZeneca Pharma; and is author of the UK and European Guidelines on Pelvic

Inflammatory Disease; is a Member of the European Sexually Transmitted Infections

Guidelines Editorial Board; is a Member of the National Institute for Health Research

Funding Committee (Health Technology Assessment programme). was previously a

Member of the National Institute for Health Research HTA Primary Care, Community

and Preventative Interventions Panel (2013-2016). He is an NIHR Journals Editor and

associate editor of Sexually Transmitted Infections journal. He is an officer of the

International Union against Sexually Transmitted Infections (treasurer), and a charity

trustee of the Sexually Transmitted Infections Research Foundation. OA and RH

declare no competing interests].

3) Please include captions for your Supporting Information files at the end of your manuscript, and update any in-text citations to match accordingly. Please see our Supporting Information guidelines for more information: http://journals.plos.org/plosone/s/supporting-information.

Reviewers' comments:

Reviewer's Responses to Questions

**Comments to the Author**

1. Is the manuscript technically sound, and do the data support the conclusions?

Reviewer #1: Yes

Reviewer #2: Yes

Reviewer #3: Partly

2. Has the statistical analysis been performed appropriately and rigorously? 

Reviewer #1: Yes

Reviewer #2: Yes

Reviewer #3: Yes

3. Have the authors made all data underlying the findings in their manuscript fully available?

Reviewer #1: Yes

Reviewer #2: Yes

Reviewer #3: Yes

4. Is the manuscript presented in an intelligible fashion and written in standard English?

Reviewer #1: Yes

Reviewer #2: Yes

Reviewer #3: Yes

5. Review Comments to the Author

Reviewer #1: I appraise the extensive systematic review for a interesting topic. The standard protocol of systematic review is well respected. The information would be helpful to readers and medical personal. Thank you.

Reviewer #2: I would like to thank the editor for the invitation to review this paper. The authors performed a systematic review and meta-analysis to evaluate the impact on different IM injection techniques and muscular pain. The paper is well-written and scientifically sound. The authors used the PRISMA criteria to extract the data and used risk assessment bias tools. While the results of this review were not conclusive and the authors acknowledge some limitations, I think this paper provide a comprehensive review about this topic.

Reviewer #3: I will focus on methods and reporting

Major

1) There are too many techniques for the number of studies, and it is difficult to provide concrete evidence.

2) relevant to the point above, it is not clear what the comparator is in all the meta-analyses reported i the abstract. "placebo" does not apply here, so this needs to be made much clearer in the abstract.

3) Publication bias tests and plots only relevant if you have >10 studies otherwise underpowered to detect much and tend to lead to conclusions that are not justified http://www.ncbi.nlm.nih.gov/pubmed/11106885. If you don’t have enough studies to assess you should discuss this as a major limitation. Even with 10 or 20 studies it is very difficult to visually assess. If you have 20 or more studies it is a considerable strength.

4) Meta-regression is a stab in the dark usually and is underpowered to detect anything but massive associations (effectively a regression with X observations, where X is the number of available studies). You should discuss this as a major limitation. Even with 60 or 80 studies, it can provide little insight.

5) Report the confidence intervals for I^2 (calculated using heterogi or metaan in Stata) as argued in http://www.ncbi.nlm.nih.gov/pubmed/17974687. A simple formula exists in the seminal 2002 Higgins paper that proposed I^2.

6) Having looked at the methods section and the studies' information, I am concerned that the comparator group is not consistent enough. The situation is not that complicated to warrant a network meta-analysis, but it seems that there is large variation - even within the "standard technique". So I am not entrirely convinced a meta-analysis like this is informative or possible, without focusing on specfic studies with identical comparison groups.

Minor

1) Abstract: add more information on methods i.e. was heterogeneity assessed and how, was publication assessed and how, random effects model used, quality of studies assessed etc.

2) Abstract: similarly add information about the results of the above in the abstract.

3) "When correlations could not be estimated, they were imputed for each outcome using the lowest (positive) estimate among other studies in the meta analysis." Is there a reference to abck this up?

4) Year may be worth considering in bias assessment, especially if you don't have enough studies for a formal test: http://www.ncbi.nlm.nih.gov/pubmed/25988604. With newer studies we would be more confident.

5) How was the random-effect model implemented, i.e. how was heterogeneity estimated? There are numerous ways to do so. Did they use the standard DerSimonian-Laird method? If so, please state so. Also there are better performing methods, for example please see https://www.ncbi.nlm.nih.gov/pubmed/28815652 (or http://www.ncbi.nlm.nih.gov/pubmed/23922860) and the metaan command in Stata where these are implemented (https://www.stata-journal.com/article.html?article=st0201).

6) Cochran Q (i.e. chi-square) is notoriously underpowered to detect heterogeneity, especially for small meta-analyses http://www.ncbi.nlm.nih.gov/pubmed/9595615. I would not use

6. PLOS authors have the option to publish the peer review history of their article (what does this mean?). If published, this will include your full peer review and any attached files.

Reviewer #1: No

Reviewer #2: No

Reviewer #3: No

---

## [Author Response · Author response to Decision Letter 0]

13 Apr 2021

Dear Editor,

We thank you for your consideration of our manuscript. The manuscript has been revised to reflect the comments from the reviewers. Please find below detailed responses to the reviewers’ comments.

Responses to reviewers’ comments

We would like to thank all the reviewers for reviewing the manuscript and for their very helpful comments and suggestions.

Reviewer #1: I appraise the extensive systematic review for a interesting topic. The standard protocol of systematic review is well respected. The information would be helpful to readers and medical personal. Thank you.

Reviewer #2: I would like to thank the editor for the invitation to review this paper. The authors performed a systematic review and meta-analysis to evaluate the impact on different IM injection techniques and muscular pain. The paper is well-written and scientifically sound. The authors used the PRISMA criteria to extract the data and used risk assessment bias tools. While the results of this review were not conclusive and the authors acknowledge some limitations, I think this paper provide a comprehensive review about this topic.

Reviewer #3: I will focus on methods and reporting

Responses to comments on methods and reporting — Reviewer #3

Major

1) There are too many techniques for the number of studies, and it is difficult to provide concrete evidence.

20 studies were meta-analysed for 4 techniques including applying local pressure to injection site (10 studies), the accupressure technique (4 studies), and Z track (2 studies), cold-needle technique (2 studies) and gluteal site injection (2 studies). Other studies and techniques were not meta-analysed either due to substantial methodological differences or because they were reported in only one study. While it may be difficult to provide concrete evidence for which technique is more beneficial in reducing intramuscular injection associated pain ( except with a network meta-analysis), that was not the focus of this study, and we aimed to review and summarise the current evidence on the effect of different intramuscular injection techniques on injection pain. 

2) relevant to the point above, it is not clear what the comparator is in all the meta-analyses reported i the abstract. "placebo" does not apply here, so this needs to be made much clearer in the abstract.

Yes, ‘placebo’ is not applicable to this study. For the meta-analyses, the comparator is ‘control’ (Lines 145-146). Clarification on the comparator for the meta-analyses has been included in the abstract, lines 28-31.

3) Publication bias tests and plots only relevant if you have >10 studies otherwise underpowered to detect much and tend to lead to conclusions that are not justified http://www.ncbi.nlm.nih.gov/pubmed/11106885. If you don’t have enough studies to assess you should discuss this as a major limitation. Even with 10 or 20 studies it is very difficult to visually assess. If you have 20 or more studies it is a considerable strength.

We agree that a higher number of studies improves the accuracy of the funnel plot and Egger’s test, but a threshold of 10 studies is widely accepted (https://doi.org/10.1002/9781119536604.ch10), and we assessed publication bias for 10 studies assessing the benefit of local pressure on injection pain. To complement our findings from the funnel plots and Egger test, we further performed a sensitivity analysis excluding small studies from the meta-analysis, and we observed a 44% reduction in the effect size (S2 Table). Therefore, we believe our assessment for small study bias to be rigorous, and findings to be robust, given the consistent trend in findings. We have also included a statement which generally reflects the limitations of publication bias testing for moderate amounts of bias or meta-analyses based on a small number of small studies (http://www.ncbi.nlm.nih.gov/pubmed/11106885) in lines 573-575. 

4) Meta-regression is a stab in the dark usually and is underpowered to detect anything but massive associations (effectively a regression with X observations, where X is the number of available studies). You should discuss this as a major limitation. Even with 60 or 80 studies, it can provide little insight.

We have included a statement discussing the limitations of meta-regressions in lines 575-579

5) Report the confidence intervals for I^2 (calculated using heterogi or metaan in Stata) as argued in http://www.ncbi.nlm.nih.gov/pubmed/17974687. A simple formula exists in the seminal 2002 Higgins paper that proposed I^2.

We have included 95% confidence intervals for the corresponding I^2 in the plots and text of the manuscript

6) Having looked at the methods section and the studies' information, I am concerned that the comparator group is not consistent enough. The situation is not that complicated to warrant a network meta-analysis, but it seems that there is large variation - even within the "standard technique". So I am not entrirely convinced a meta-analysis like this is informative or possible, without focusing on specfic studies with identical comparison groups.

We reviewed the description of the comparator and sought further information from the study authors where possible. The comparators were mostly control and standard of care, and it was only when the standard of care was adjudged to be clinically different (for instance, when the comparator involved a different technique, rather than a control) that it was excluded from the meta-analysis, Lines 145-146. Nuances in the delivery of intramuscular injections are common, and can reflect local or national guidelines. Therefore, we adopted a pragmatic approach by including these studies, excluding them based on the variations in the standard of care may introduce limitations to the generalisation of the findings. We further express the likely impact of these nuances in our conclusions, “but any conclusions are limited by small study sizes, non-standardised interventions, imprecise control groups and wide inter-study heterogeneity.”—lines 585-587

Minor

1) Abstract: add more information on methods i.e. was heterogeneity assessed and how, was publication assessed and how, random effects model used, quality of studies assessed etc.

The word count for the abstract is limited, but we have made a concise summary of the methods, lines 25 -29, “The review protocol was registered on PROSPERO (CRD42019136097). MEDLINE, EMBASE, British Nursing Index and CINAHL were searched up to June 2020. Included studies were appraised and a meta-analysis, where appropriate was conducted with a random effects model and test for heterogeneity. Standardised mean difference (SMD) with a 95% confidence interval in reported injection pain (intervention cf. control) was reported”. 

2) Abstract: similarly add information about the results of the above in the abstract.

SMDs and 95%CI for the techniques are reported in the abstract, lines (34-44). Comments of the quality of studies, heterogeneity are made in Included in lines 44-46, “Limitations included considerable heterogeneity, poor reporting of randomisation, and possible bias in outcome measures from unblinding of assessors or participants.”

3) "When correlations could not be estimated, they were imputed for each outcome using the lowest (positive) estimate among other studies in the meta analysis." Is there a reference to abck this up? 

Yes, the references https://doi.org/10.1093/ije/31.1.140 and DOI: 10.1002/14651858.CD002066, have been cited in the manuscript

4) Year may be worth considering in bias assessment, especially if you don't have enough studies for a formal test: http://www.ncbi.nlm.nih.gov/pubmed/25988604. With newer studies we would be more confident.

Most of the identified studies (22/29) were published within the last decade (after 2011), and 16/20 studies meta-analysed. While findings and methodologies from older studies may be less robust, the risk of such bias given the small number of older studies identified may be low. More so, the quality of the reported methodology and findings for all the included studies was assessed in Figure 2 and Table 2.

5) How was the random-effect model implemented, i.e. how was heterogeneity estimated? There are numerous ways to do so. Did they use the standard DerSimonian-Laird method? If so, please state so. Also there are better performing methods, for example please see https://www.ncbi.nlm.nih.gov/pubmed/28815652 (or http://www.ncbi.nlm.nih.gov/pubmed/23922860) and the metaan command in Stata where these are implemented (https://www.stata-journal.com/article.html?article=st0201).

The standard DerSimonian-Laird method was used and has been mentioned in the manuscript, Lines 165-166

6) Cochran Q (i.e. chi-square) is notoriously underpowered to detect heterogeneity, especially for small meta-analyses http://www.ncbi.nlm.nih.gov/pubmed/9595615. I would not use

We have removed the Cochran Q heterogeneity assessment and reported I^2 with 95%CI

---

## [Editor Report · Decision Letter 1]

16 Apr 2021

The effect of intramuscular injection technique on injection associated pain; a systematic review and meta-analysis

PONE-D-20-24754R1

Dear Dr. Oluseyi Cyril Ayinde

We’re pleased to inform you that your manuscripts have been judged scientifically suitable for publication and will be formally accepted for publication once it meets all outstanding technical requirements.

Kind regards,

Ma Angeles Peña

Academic Editor PLOS ONE

---

## [Editor Report · Acceptance letter]

23 Apr 2021

PONE-D-20-24754R1 

The effect of intramuscular injection technique on injection associated pain; a systematic review and meta-analysis 

Dear Dr. Ayinde:

I'm pleased to inform you that your manuscript has been deemed suitable for publication in PLOS ONE. Congratulations! Your manuscript is now with our production department. 

Kind regards, 

on behalf of

Dr. Mª Ángeles Peña Fernández 

Academic Editor

PLOS ONE